# Recent Insight on Edible Insect Protein: Extraction, Functional Properties, Allergenicity, Bioactivity, and Applications

**DOI:** 10.3390/foods11192931

**Published:** 2022-09-20

**Authors:** Jiayin Pan, Haining Xu, Yu Cheng, Benjamin Kumah Mintah, Mokhtar Dabbour, Fan Yang, Wen Chen, Zhaoli Zhang, Chunhua Dai, Ronghai He, Haile Ma

**Affiliations:** 1School of Food and Biological Engineering, Jiangsu University, 301 Xuefu Road, Zhenjiang 212013, China; 2Institute of Food Physical Processing, Jiangsu University, 301 Xuefu Road, Zhenjiang 212013, China; 3CSIR–Food Research Institute, Accra P.O. Box M20, Ghana; 4Department of Agricultural and Biosystems Engineering, Faculty of Agriculture, Benha University, Qaluobia P.O. Box 13736, Egypt; 5School of Food Science and Engineering, Yangzhou University, 196 Huayang West Road, Yangzhou 225127, China

**Keywords:** insect protein, extraction, functional characteristics, allergenicity, biological activity, ace-inhibition activity

## Abstract

Due to the recent increase in the human population and the associated shortage of protein resources, it is necessary to find new, sustainable, and natural protein resources from invertebrates (such as insects) and underutilized plants. In most cases, compared to plants (e.g., grains and legumes) and animals (e.g., fish, beef, chicken, lamb, and pork), insect proteins are high in quality in terms of their nutritional value, total protein content, and essential amino acid composition. This review evaluates the recent state of insects as an alternative protein source from production to application; more specifically, it introduces in detail the latest advances in the protein extraction process. As an alternative source of protein in food formulations, the functional characteristics of edible insect protein are comprehensively presented, and the risk of allergy associated with insect protein is also discussed. The biological activity of protein hydrolyzates from different species of insects (*Bombyx mori*, *Hermetia illucens*, *Acheta domesticus*, *Tenebrio molitor*) are also reviewed, and the hydrolysates (bioactive peptides) are found to have either antihypertensive, antioxidant, antidiabetic, and antimicrobial activity. Finally, the use of edible insect protein in various food applications is presented.

## 1. Introduction

The world’s population in 2022 is about 6.9 billion people greater than it was in 1800. With the increasing human population, protein shortage has become a global problem [1]. One possible sustainable alternative, in addressing this problem, is to use protein from edible insects [2] and underutilized plants. For centuries, edible insects have been important sources of nutrients for human nutrition around the world. More than 2100 insect species have been listed as edible in many continents, including Asia, Africa, and Latin America [1,3]. Recent advocacy by the United Nations’ Food and Agriculture Organization (FAO) on the exploitation of insect resources has made the exploitation of insects as food or food ingredients a new research focus in the food sciences [1]. Apart from the benefits to food production, insect farming could significantly reduce the current global use of land and water, and produce fewer greenhouse gas emissions than animals such as cattle or pigs [4,5]. The sustainability of insect farming makes insects one of the potential alternative sources of protein to address the global protein shortage [3,6].

Although the nutritional composition of insects varies considerably depending on species, stage of development, environmental conditions, and feed [7], they are generally considered an important source of high-quality protein, with a balanced amino acid content, essential fatty acids, microelements, and other bioactive compounds [8]. The nutritional compositions of some common edible insects are shown in Table 1. Most insects are high in micronutrients such as potassium, calcium, iron, magnesium, and selenium. Insects contain more iron and calcium than beef, pork, and chicken [1]. In most cases, when compared with plant and meat protein, insects have higher quality proteins in terms of their nutritional value, total protein content, essential amino acid composition, and protein efficiency (from feed) [9,10]. This information is important, and may encourage the acceptance of insects in the diets of many worldwide. Essential amino acid contents of edible insects are shown in Table 2.

Consumer acceptance of insects, however, remains low in Western countries, largely due to cultural habits and bias [14]. Nevertheless, when insects are used in powdered form as ingredients in various foods to conceal the insects’ appearance, individuals are more receptive to them [15]. Hence, it is effective to extract protein from insects and provide insects to human consumers in the form of powder, meal, or fraction. However, in order to incorporate insects into different foods, their functional properties must be thoroughly evaluated following various forms of treatments and processing [16]. A major concern regarding consumption of insects is possible allergenic responses [17]. Indeed, insects have been implicated in cross-allergenic reactions to crustacean and house dust mite proteins [18,19]. Insects have great biodiversity and biomass, potentially providing a great natural resource for the development of new bioactive peptides [20]. Several studies demonstrate that insect peptides/polypeptides have antihypertensive, antioxidant, antidiabetic, and antimicrobial activities [1,17,21,22]. Mass production of insect bioactive peptides is a promising biotechnology business.

As a future trend, the consumption of insect protein is a valuable means not only to ensure food security in areas with significant population growth and scarce resources, but also to improve the sustainability of food production with high nutritional value. Motivated by such a background, this work aims to provide a comprehensive and systematic review on the extraction, functional properties, allergenicity, bioactivity, and applications of protein and hydrolysate from some common edible insects (*Bombyx mori*, *Hermetia illucens*, *Acheta domesticus*, *Tenebrio molitor*), to provide researchers, entrepreneurs, and policymakers with extensive information on the subject.

## 2. Extraction of Insect Protein

Insect protein is usually extracted from skim insect powder. Protein extraction methods are generally classified into conventional (aqua-based, salt, solvent, detergent, alkali) and non-conventional, or advanced, green extraction methods (enzyme, ultrasound, microwave, and pulsed electric field assisted extraction) [23]. Among them, the most common method for isolating protein fractions is alkaline extraction. Insect proteins can selectively solubilize at alkaline pH (i.e., about 70% crude protein). During the dissolution process, other undesirable components such as fat and exoskeleton could be removed, followed by precipitation of insect protein at pH 4–5. This process ultimately results in a purified and concentrated insect protein [24]. The solubility of proteins increases with an increase in pH of the solvent due to the ionization of acidic and neutral amino acids at high pH. Therefore, the extraction of protein in an alkaline environment results in higher protein yields [25]. Furthermore, several reports reveal that the protein extraction rate of insects increases with increasing pH [24,26,27,28]. However, it was reported that the extractability of cricket protein using ascorbic acid at low pH (2 and 3) was higher than that using sodium hydroxide at pH 13 [29]. Mintah et al. [30] used alkali extraction of protein (0.25 M NaOH) for microwave-dried *Hermetia illucens* and found that the highest extraction rate (64.44%) was noticed at extraction time 59.43 min, alkaline solution to sample ratio 24.85:1.00, and temperature 52.23 °C. The protein content of edible insects was estimated by the Kjeldahl method, using the nitrogen (N) to protein conversion factor (6.25). Recently, however, Levia and Martinez [31] reported that this factor may not be suitable for insects, because the insect cuticle contains a large amount of fibrous chitin, N-rich polysaccharides, and proteins tightly embedded in its matrix, which would render the estimated protein content too high. Therefore, the total N and the total non-digestible N multiplied by the conversion factor were calculated (Figure 1). This approach consumes more time, but should replace the converted N-content evaluation methods for quantifying an insect protein content that is nutritious for humans and animals. Janssen et al. [32] determined a conversion factor of 4.76 for whole larvae from *Tenebrio molitor*, *Alphitobius diaperinus*, and *Hermetia illucens*, using amino acid analysis. The edible insects were subsequently subjected to an aqueous extraction protocol and dialysis through a 12–14 kDa molecular weight cut-off membrane to collect water-soluble proteins. The conversion factor of the water-soluble protein extracts of *T. molitor*, *A. diaperinus*, and *H. illucens* larvae was estimated to be 5.60.

Compared with traditional organic solvents, the use of ionic liquids (ILs) to isolate proteins can effectively maintain the biological activity and stability of proteins. Therefore, Zeng et al. [33] used choline hydroxide ionic liquid (CH-IL) as the extraction solvent to extract silkworm pupa protein and discovered that the protein content increased by 12.14% compared with the traditional alkaline solubility-acid precipitation method. Zhang et al. [34] noted that using 1-ethyl-3-methylimidazolium chloride as solvent, dissolving at 90 °C for 24 h, and ethanol bath as regeneration solvent, the highest yield of silkworm pupa protein was 62.6%, and the low-fat content was less than 0.5%. The use of salting-assisted extraction is a well-established technique that can enhance alkaline solubility-acid precipitation (AEAP) extraction efficiency and preserves protein functionality. Jiang et al. [35] studied protein extraction of *Tenebrio molitor* larvae using the AEAP assisted by NaCl (salting-in) and (NH_4_)_2_SO_4_ (salting-out) methods. AEAP extraction was conducted by dispersing the larvae powder in 1.5% NaOH solution and then conducting precipitation of the supernatant at pH 4.5. Salting-in-AEAP extraction was similar to the AEAP extraction method, except that 1% NaCl (*w*/*v*) was added to the NaOH solution. The AEAP-salting-out extraction was similar to AEAP extraction; only 20% of (NH_4_)_2_SO_4_ was added to the supernatant to accelerate the protein precipitation. Salting-in-AEAP-out extraction was conducted by adding 1% of NaCl (*w*/*v*) to the NaOH solution and adding 20% of (NH_4_)_2_SO_4_ to the supernatant. The protein extraction yield of simple AEAP extraction was 22.26% protein, while the yield increased to 28.3, 37.5, and 39.5%, by salting-in assisted treatment, AEAP-salting-out extraction, and salting-in-AEAP-out extraction, respectively.

Conventional methods refer to the commonly used techniques which may sometimes result in lower extraction yields due to protein degradation. This reduction in protein yield is influenced by a variety of factors, including extraction time, solvents, pH, and temperature. Therefore, researchers are currently more focused on advanced green technologies to improve extractability and reduce protein degradation during extraction. Enzymatic protein extraction was rarely used for the high-yield extraction of insect proteins due to its slow, expensive operation (due to the exogenous price of enzymes), difficulty expanding the extraction scale, unstable yield, and other shortcomings. However, the use of enzymatic hydrolysis to obtain bioactive protein hydrolysates in insects has been widely studied. For some physical extraction methods (ultrasound, microwave, and pulsed electric field) combined with conventional protein extraction methods or other non-conventional extraction methods (physical or enzymatic), the efficiency of protein extraction can be significantly improved [36,37]. First, ultrasound was the most widely used technology in assisted extraction. Ultrasonic processing is considered a green alternative technique because it is simple to operate, time-efficient, and requires less solvent [38]. On the other hand, microwave technology has some advantages in terms of its green and sustainable parameters, such as mild temperature and shorter extraction time, but it generates a large amount of thermal energy, leading to the degradation of thermally unstable bioactive compounds [39,40]. Compared with other unconventional methods, pulsed electric fields are relatively inefficient in obtaining higher protein yields. The use of pulsed electric field extraction at lower temperatures, longer pulse durations, and higher field intensities resulted in more recovery of the proteins [23]. Psarianos et al. [36] found that cricket protein yield increased by 32.47% after 15 min extraction, and 18.62% after 60 min extraction following pulsed electric field treatment (4.90 kJ/kg). In addition, after 15 min of extraction, the extraction yield of 24.53 kJ/kg and 49.10 kJ/kg treatments was 30.47 and 39.55% higher than that of the control, respectively.

## 3. Functional Characteristics of Insect Proteins

A thorough understanding of the functional characteristics of insect protein can facilitate the successful application of such protein in various food systems. Insect processing technology and protein extraction directly affect the functional characteristics of insect protein [41,42]. The changes in the functional properties of proteins are mostly linked to alteration of surface charge, surface hydrophobicity, and molecular weight distribution of proteins [37]. After defatting, insect powder may stimulate protein aggregation, resulting in a decrease in the hydrophobicity of isolated protein and consequently causing the reduction of emulsifying property [10]. After the extraction of insect protein, the chitin content was significantly reduced, but the foaming ability and stability of the protein were noticeably improved [10]. Insect protein and chitin may both have emulsifying properties. Therefore, to determine whether the emulsifying properties came from protein or chitin particles, the interface and emulsifying properties of cricket powder were evaluated. Results indicated that both water-soluble protein and milled chitin particles in cricket powder could help stabilize the oil-in-water emulsion and form a stable emulsion fraction [43]. However, only the smaller particle sizing of milled chitin was useful for emulsification, limiting their relevance as industrial emulsifiers [43].

The functional characteristics of different insect species are different. Compared to *Tenebrio molitor*, the *Protaetia brevitarsis* and *Allo-Myrina dichotoma* edible insects exhibit higher protein thermal stability and emulsification activity. Therefore, they may possess better manufacturing characteristics [44]. The higher concentration of polyphenols (mainly epicatechin gallate) in the protein concentrate of *Hermetia illucens* may lead to more interaction between protein and polyphenols, altering the interaction of amino acids in the oil phase, and consequently reducing emulsification of *Hermetia illucens* protein [45].

The pH value, ionic strength, and temperature during protein extraction may influence the functionality of insect protein. Heat-induced soluble protein aggregation is dependent on temperature and pH, and variations in protein size distribution is dependent on the same factors [46]. Purschke et al. [47] examined the effects of different pH values (2−10) and salt concentrations (0, 1, and 3% *w*/*v*) on the functional characteristics of migratory locust (*Locusta migratoria*) protein. The maximum solubility (100%) of migratory locust protein was observed at pH 9. The emulsifying activity of migratory locust protein at pH 5, foaming ability at pH 3 and 3% NaCl, and foam stability at pH 9 could reach the same level as egg white protein. The foamability (after heat treatment −75 and 95 °C, 15 min) and the critical gelation concentration (at 6.5% *w*/*w*) of the black cricket protein could also reach the same level as whey protein isolates [48]. Queiroz et al. [49] investigated the foam properties of protein extracts from *Hermetia illucens* larvae treated at different temperatures (75 and 85 °C for 15 min) and found that the optimal foam stability was achieved at 85 °C for 15 min. The solubility, emulsification, foaming ability, and stability of extracted protein were significantly improved by the alkaline extraction and acid precipitation methods assisted by NaCl (salting-in) and (NH_4_)_2_SO_4_ (salting-out) procedures [35].

Insect proteins generally have good oil and water retention capacity, emulsion activity, and foaming activity [50]. Among them, edible grasshopper (*Schistocerca Gregaria*) and honeybee (*Apis mellifera*) display highly emulsifying properties comparable to whey protein, showing their potential as alternative emulsifiers [37]. Protein extracts from *Hermetia illucens* (obtained by conventional means), like other plant and insect species, have poor solubility in water and other aqueous media, which may affect their nutritional value, biological activity, techno-functionality, and applications in food formulations [51]. Therefore, to better use insect protein in food, the functional characteristics of insect protein must be improved. Recently, some new modification techniques have been adopted, such as ultrasound [37,42], ultra-high pressure, pH-shifting technology [52], and cold atmospheric pressure plasma processing [53], etc. The emulsification and foaming ability of insect protein were improved after ultrasonic treatment [37]. Kumar et al. [42] studied the effect of ultrasonication time (5,15 and 30 min) on the gelation of the protein of *Hermetia illucens* larvae and observed that the gel strength became stronger after sonication for 15 min, and the gel had the densest microstructure and the lowest pore size, indicating a good gel system. The quality of myofibrillar protein was significantly improved by combining the insect protein extracted by acidic pH-shifting technology with myofibrillar protein [52]. When cold atmospheric pressure plasma technology was used to modify *Tenebrio molitor* protein, the oil-binding capacity increased from 0.59 to 0.66 g/g with increased exposure to cold atmospheric pressure plasma [53]. On the other hand, enzymatic modification of proteins is an effective way to observably improve the functional properties of insect proteins [54]. Hall et al. [54] found that enzymolysis (using alcalase) considerably improved the solubility of cricket protein under varying pH (3, 7, 8, and 10), and it also enhanced emulsibility and foaming under low acid. With this context, the application of insect protein modification technology should be determined according to the insect species and the functional characteristics required by the application.

## 4. Allergenic Risks of Insect Protein

There are many questions about the safety of using insects as food, which involve the following three risks: biological, chemical, and allergenic [16]. For biological risks, viruses that are pathogenic for insects are not risks for humans because they are specific to invertebrates, and due to the genetic difference between humans and insects; as well, as processing of the insects (e.g., freezing and cooking) can help eliminate this risk [55]. According to the European Food Safety Authority (EFSA), the chemical contaminants of greatest concern are heavy metals such as cadmium, mercury, lead, and arsenic, as well as the accumulated pollutants from the environment such as hormones and pesticides [16]. The problem of chemical contaminants can be addressed from the perspective of feeding the insects, aiming at preventing and minimizing the accumulation of toxins, drugs, and antinutrients from the external environment. One of the most critical steps is to control chemical risks at the breeding stage and transformation process [55]. Although biological and chemical risks have been reported, the main limitation to the widespread use of insect proteins is the risk of allergies.

The literature confirms the possibility of humans developing food allergies to insects, and that the risk of anaphylaxis from insect consumption is significant [55]. A variety of potential allergens, such as 27-kDa glycoprotein, Bom m 9, thiol peroxiredoxin, chitinase, and paramyosin, have been found in silkworm pupae, which can bind IgE in the serum of silkworm allergic patients [56]. Proteomic data revealed that the expression of certain proteins in the pupae of silkworm larvae varied with sex and feeding habits. Female silkworm pupae reared on mulberry leaves had lower levels of known allergens than under other conditions [56]. Moreover, allergen abundance and detectability vary with extraction methods and food processing techniques [57].

The adverse reactions described following the consumption of insects, including anaphylactic shock, can be attributed to a primary allergic sensitization mechanism or, more likely, IgE cross-reactions in sensitized species related to other classifications of the arthropod genus [58]. Common allergens in various insects include tropomyosin [59,60] and arginine kinases [60,61], both of which are pan-allergens known for their cross-reactions with homologous proteins of crustaceans and dust mites [62]. Marchi et al. [58] studied the allergenic relevance of crickets through cross-reaction with Pacific white shrimp *Litopenaeus vannamei*. The results revealed that tropomyosin was identified as the most significant IgE binding protein, and its cross-reactivity of tropomyosin and shrimp tropomyosin was verified by ELISA. Silkworm bodies contain allergens in their different growth stages (larva, pupa, moth) and their metabolites (silk, molting, feces). A total of 45 potential allergens were found in silkworms, among which 7, 16, 17, 4, 3, and 4 allergens were detected in larvae, pupae, moths, silk, slough, and feces, respectively [63]. The results of the homology comparison indicated that the allergens of silkworms were likely to cross-react with the allergens of *Dermatophagoides farinae*, *Aedes aegypti*, *Tyrophagus putrescentiae*, *Triticum aestivum*, and *Malassezia furfur* [63]. Based on phylogenetic relationships, the frequently reported IgE-binding cross-reactions between these allergens from different sources may trigger some unexpected cross-allergic reactions in susceptible individuals [64]. Due to the close taxonomic relationship between arthropods and crustaceans, patients allergic to house dust mites, shrimp, and mealworms should be cautious in eating insects [55,65]. The correct labeling of food containing insects can help protect the health of allergic consumers [66].

The clinical manifestations of insect food allergies range from mild local reaction to severe systemic clinical manifestations such as anaphylactic shock. Reported symptoms can be divided into skin (e.g., hives, itching, rash, flushing, angioedema), gastrointestinal (e.g., abdominal pain, nausea, vomiting, diarrhea), and respiratory (e.g., asthma, dyspnea) [62]. Moreover, the duration of symptoms can range from a few minutes to 6 h [62]. Therefore, many effective strategies have been proposed to reduce the sensitization of insect proteins, such as heating, microwaving, glycosylation, high pressurization and enzymatic hydrolysis [19,22,56,67]. Heat treatment may reduce this risk, but does not eliminate it [68].

Microwave-assisted enzymatic hydrolysis is an effective method used to prepare bioactive peptides from insect proteins to reduce their immunoreactivity [22]. Raman spectra showed conformational changes, especially in the Amide I and S-S regions, which may be related to the observed immunochemical reactions [22]. Further, heat, enzymolysis, and acid-alkali treatment can significantly decrease the allergenicity of silkworm pupa protein, with heat-, enzyme- and acid-alkali–resistant allergens at 25 to 33 kDa silkworm pupa protein [56]. No potential allergic reactions were observed when Sprague-Dawley rats were fed freeze-dried *Tenebrio molitor* up to 3000 mg/kg/day for 90 days [69]. Compared with unhydrolyzed crickets, most of the cricket protein hydrolysates in shrimp allergy serum have lower tropomyosin IgE reactivity [17]. However, cricket protein must be hydrolyzed to a degree of 60–85% to completely eliminate the IgE response to tropomyosin [17]. Interestingly, cricket protein hydrolysate (52% degree of hydrolysis) displayed similar reactivity to the unhydrolyzed sample, and thus requires further investigation [17].

## 5. Bioactivity of Insect Proteins

### 5.1. Silkworm Pupa (Bombyx mori)

Medical studies have shown that consumption of silkworm pupae protein may be associated with a reduced risk of many diseases, such as cardiovascular diseases (hypertension and hyperlipidemia) [70,71,72,73,74,75], and cancer (hepatoma and gastric cancer) [76,77]. Additionally, enzymatically produced silkworm pupa hydrolysate is reported to possess several bioactive properties, such as inhibiting angiotensin-I converting enzyme (ACE) activity [70,71,72,73,74], antioxidant activity [78,79,80], immunomodulatory activity [81], improving hypercholesterolemia [75], and anti-tumor activity [76,77]. Such biological activities are mostly due to the formation of strong hydrogen bonds [71,75]. After enzymatic hydrolysis, not only does the allergenicity of silkworm pupa protein decrease [56], but in addition, umami peptide could be separated from the protein hydrolysate of silkworm pupae [82]. Therefore, protein hydrolysates from silkworm pupae may not only serve as an important source of dietary protein, but may also provide additional benefits for human health [77].

ACE plays an important role in the regulation of blood pressure and is considered a key method in the treatment of hypertension [70]. Antioxidants may prevent or treat reactive oxygen species (ROS)-related human disease (cancer, diabetes, atherosclerosis, arthritis, and neurodegenerative diseases) [78]. However, synthetic ACE inhibitors/antioxidants have been suspected of impacting health [73,78]. Therefore, there is increasing interest in ACE inhibitors and antioxidants from natural products. Several protease hydrolysates with ACE inhibitory activity, such as neutral protease hydrolysates, acid protease hydrolysates, flavor protease hydrolysates, trypsin hydrolysates, and gastrointestinal protease hydrolysates, have been reported for silkworm pupa protein [70]. The area and shape of the infrared absorption peak of silkworm pupa protein were significantly changed after enzymatic hydrolysis, and the size of the infrared absorption peak was greatly reduced with the increase of the number of cracks; as well, the content of sulfhydryl groups was significantly increased, but the concentration of disulfide bonds was significantly decreased [79]. These structural changes ultimately resulted in the enhancement of antioxidant and ACE-inhibition activity [79]. Tao et al. [71] isolated and purified a peptide with ACE-inhibition activity (GLy-ASN-Pro-TrP-Met (GNPWM)) from silkworm pupae protein, and synthesized four modified peptides (GNPWW, NPWW, PWW, and WW) derived from this peptide. Among these modified peptides, WW exhibited the highest ACE-inhibition activity by hydrogen bonding with His383 and distortion of the tetrahedrally-coordinated Zn (II), resulting from strong hydrogen bonds between peptides with some critical ACE residues and coordination interactions with Zn (II). When the protein components in silkworm pupae protein were separated and hydrolyzed (albumin, globulin, glutelin, and prolamine), it was noted that albumin was the most easily hydrolyzed component, and the degree of hydrolysis and ACE-inhibition rate were higher than the other components [74]. Furthermore, the ACE-inhibition activity differed according to the method used, which requires a standardized technique to evaluate it [21]. Cermeño et al. [80] hydrolyzed silkworm pupae protein using different enzymes, i.e., Alcalase^®^ (2.4 Anson Units (AU) g^−1^; (Sigma-Merck, Dublin, Ireland)), Prolyve^®^ (2.2 AU g^−1^; (Lyven Enzymes Industrielles, Caen, France)), Flavourzyme^®^ (500 Leucine Amino Peptidase Units g^−1^; (Sigma-Merck, Dublin, Ireland)) and Brewers Clarex^®^ (37 AU g^−1^; (DSM, Kaiseraugst, Switzerland)), and observed that using Alcalase and Prolyve resulted in hydrolysates with high scavenging activity. Hydrolysates produced with Flavourzyme and Brewers Clarex exhibited higher iron-reducing antioxidant capacity than other samples. On the other hand, the pupal protein of two natural silkworm strains, Nangnoi and Nangsew, were hydrolyzed using alkaline protease (Alcalase), papain, and trypsin. The findings showed that the hydrolysate of the Nangnoi strain prepared by alkaline protease had the highest antioxidant activity [78].

Immunomodulatory peptides have positive effects on body function, but many immunomodulators are not suitable for long-term use or preventive use [81]. Therefore, the identification of new immunomodulators to enhance non-specific host defense mechanisms is an ongoing research direction [81]. It is an effective method to obtain immunomodulatory peptides from insect protease hydrolysates. A new immunoregulatory hexapeptide (Pro-ASN-Pro-ASN-THR-ASN) was purified from the enzymatically produced silkworm pupae protein hydrolysate [81]. In addition, purified peptides in the presence of Concanavalin A or lipopoly-saccharide could significantly stimulate the proliferation of T-lymphocytes and/or B-lymphocytes, which are stable to high temperature and gastrointestinal protease [81]. Therefore, this novel immunomodulatory hexapeptide has potential therapeutic value as an immunomodulatory component of functional food. Some target therapeutic drugs have been used to treat hyperlipidemia with various side effects, such as gastrointestinal discomfort, myopathy, and hepatotoxicity. However, Sun et al. [75] found that the hydrolysis products of silkworm pupae protein (His-Pro-Pro (HPP) and Ser-Gly-Gln-ArG (SGQR)) under neutral proteinase could improve hypercholesterolemia, and the effect on inhibiting cholesterol biosynthesis was mainly due to the formation of strong hydrogen bonds. In vitro studies showed that HPP and SGQR inhibited the activity of high 3-hydroxy-3-methyl glutaryl coenzyme A reductase (HMGCR), decreased the mRNA and protein expression of HMGCR and squalene synthase, and activated the expression of low-density lipoprotein receptors [75].

Selenium-rich amino acids were obtained from silkworm pupae protein by hydrolysis using trypsin and papain. Studies have demonstrated that selenium-rich amino acids could increase the production of intracellular ROS concentration-dependently and inhibit hepatocellular carcinoma by inducing apoptosis through ROS production [76]. The antitumor activity of silkworm pupae protein hydrolysates has been studied [70], and the results suggested that they can significantly inhibit the proliferation of several malignant tumor cells, including gastric cancer cells [77]. Although the proapoptotic properties of protein hydrolysates from silkworm pupae have been previously confirmed, the specific apoptosis pathway and activation mechanism are however still unclear. Li et al. [77] studied the proapoptotic effect of silkworm pupae protein hydrolysates (SPPH III, peptide < 3 kDa) on MGC-803 gastric cancer cells in vitro. Figure 2 illustrates the rationale for the proposed mechanism of action. The antitumor activity of SPPH III can be attributed to its inhibition of MGC-803 cell viability and promotion of apoptosis. Inhibition of Bcl-2 expression results in the destruction of mitochondrial membrane potential (δ ψ M), and the activation of p53 and caspase-9 results in the activation of mitochondrial apoptosis pathways. Therefore, the results suggest that SPPH III is a potential drug candidate for the treatment of gastric tumors.

To better obtain bioactive peptides from insect protein hydrolysate, some new techniques, such as ultrasound, have been applied to assist enzymatic hydrolysis [83]. Jia et al. [83] demonstrated that ultrasonic pretreatment was a successful means for improving the ACE-inhibition activity of silkworm pupae protein hydrolysate. New metal organic frameworks material with specific recognition and capture capability is available to enable efficient and selective extraction of antioxidant peptides from silkworm pupa protein hydrolysate-N [84]. He et al. [84] used Histidine (His), which is widely presented in antioxidant peptides, to construct the modified material ZIF-8 (ZIF-His) with defective mesopore imprinting based on the “molecular imprinting strategy”. Then, ZIF-His was used to have a high adsorption capacity for antioxidized peptides, especially for peptides containing His, tryptophan, phenylalanine, and tyrosine. Finally, three peptides with the highest antioxidant activity (FKVPNMY, AVNMVPFPR, and VNMVPFPR) were identified, and had excellent free radical scavenging activities and Ferric reduction abilities.

### 5.2. Black Soldier Fly Larvae (Hermetia illucens)

*Hermetia illucens* (*H. illucens*) is an edible insect that provides both manifest (i.e., nutrient) and latent benefits (e.g., waste degradation) to humans [51]. *H. illucens* are rich in fat and protein, among other components (fiber, minerals, vitamins), making them a good source for food fortification [52]. The nutrient composition of *H. illucens* varies among substrates (feed) not only in protein content (ranging from 37 to 63% dry matter) but also fat content, which has the most variation (ranging from 7 to 39% dry matter) [85]. Bioactive compounds/peptides can be produced during gastrointestinal digestion [86]. The antioxidant activity of edible insects after simulated gastrointestinal digestion is higher than that of some protein hydrolysates obtained from plants or other animal products [86]. However, other studies suggest that insect cell extracts have in vivo antihypertensive activities that do not require additional digestion by gastrointestinal enzymes [87]. Bioactive compounds/peptides can also be produced by in vitro proteolysis to prepare small molecular weight compounds with many human health benefits, such as antioxidant, antihypertensive, and anti-inflammatory effects [88]. Therefore, protein hydrolysates are better substitutes for complete protein and/or element formulations.

The preparation of bioactive hydrolysates/peptides (from proteins) by proteolysis is considered to be most appropriate method compared to the use of acids or alkali [88]. The reaction conditions are mild and controllable, the product quality is high, and the protease is available on the market [88]. Factors such as hydrolysis time, temperature, pH, and enzyme and substrate concentration influence the enzymatic hydrolysis process (in vitro) and thus the bioactivity of proteolytic products [88]. The function of *H. illucens* protein may be improved after hydrolysis. The properties of protein hydrolysates depend on molecular size and amino acid composition/sequence that shapes their structure [89]. Following enzymatic hydrolysis, protein extracts can undergo significant structural changes and improve their biological activities. Firmansyah et al. [89] optimized the proteolysis process of *H. illicens* protein (using the central composite design method). The proteolysis process was conducted under varying enzyme (bromelain) concentrations (1–5%), pH values (6–8), and time (3–24 h). The mathematical model revealed that the optimal enzymolysis conditions were as follows: enzyme concentration 3%, pH 8 and time 24 h, and maximal degree of hydrolysis (DH) 47.4%. The protein hydrolysate had a yield of 10.70% (on a weight basis) based on defatted biomass with a productivity of 21 mg/L/batch. The obtained *H. illicens* protein hydrolysate contained a high content of hydrophobic essential amino acids, especially lysine (8.0%), leucine (7.7%), and valine (7.3%). At 1.25% of the sample concentration, *H. illicens* protein hydrolysate could scavenge 77% of DPPH (2,2-diphenyl-2-picrylhydrazyl) free radical, with an IC_50_ value of 0.84% *v*/*v*.

Traditional enzymolysis is also thought to limit the enzymatic hydrolysis of proteins due to the incompatible conformation of protein, making it difficult for proteases to attack the protein’s enzyme-driven bonds [51]. To overcome this limitation, ultrasonication of samples prior to enzymolysis could be useful in preparing hydrolysates with enhanced bioactivity and antioxidant capacity [51,88]. This is attributed to the cavitation effects of ultrasound (acoustic properties), which results in strong shear thrust, turbulence, and shockwaves [88]. Mintah et al. [51] discovered that different pretreatment methods (conventional, ultrasonic-fixed, and ultrasonic-sweep frequency) of *H. illucens* could be used to prepare proteins and/or hydrolysates and enhance their techno-functionality and antioxidant capacity. They found that the solubility, foam, and antioxidant capacity (ABTS (2,2′-Azinobis-(3-ethylbenzthiazoline-6-sulphonate))), superoxide scavenging, and Ferric-reducing) of the hydrolysate treated by ultrasonic wave (especially sweep frequency) were better than those of other modes and isolates (*p* < 0.05). Treatment type affected the microstructure, functional properties, and antioxidant capacity. Therefore, functional and antioxidant properties could be improved or modified for different food applications based on selective processing. Moreover, Mintah et al. [90] studied the structural attributes of the above-mentioned ultrasonically pretreated *H. illucens* protein isolates and hydrolysates and noted that the thiol value, dispersibility (pH 2–10), and surface charge of the isolate and hydrolysate treated by ultrasonic wave (especially sweep-type) were significantly increased, while the turbidity and the particle size of hydrolysate and protein isolate substantially decreased. At the same time, the secondary structural components were changed. Furthermore, Mintah et al. [88] observed that sonicated *H. illucens protein* hydrolysates showed higher oxidation activity (ion chelating activity, DPPH-radical scavenging activity, Hydroxyl radical scavenging activity, and cupric ion chelating activity) relative to nonsonicated samples.

### 5.3. Cricket (Acheta domesticus)

Due to their high protein content and nutritional value, edible insects, especially crickets, are currently considered solutions to the growing protein demand worldwide [17]. Crickets are permitted by European legislation and have high nutritional value [91]. Crickets are rich in protein (55–60%), fat (24–29%), fiber (3.5–7%), and minerals [92]. Regarding mineral content, crickets are rich in Ca, Mg, and Fe, especially the content of Cu, Mn, and Zn [92]. Four cricket-specific peptides were found in the trypsin lysates of cricket powder, and these peptides showed sufficient thermal stability [92]. In addition, previous studies suggested that the high levels of chitin and chitosan found in cricket inhibits intestinal pathogens, but high levels of chitin reduce the functional properties of cricket powder as a food ingredient [91]. Therefore, the hydrolysis of cricket meal could be an innovative strategy to improve the sensory and functional properties of insect proteins [91].

The processing of insects, including the type of heat treatment (boiling, cooking, baking), also significantly affected the nutritional value and biological activity of edible insects [92]. Changes in processed cricket flour are attributed to thermally induced phenomena such as protein denaturation, cross-linking, Maillard glycosylation, and/or aggregation [93]. Thermal processing of cricket powder can alter its function and improve its proton-providing antioxidant function [93]. This increased antioxidant activity may be attributed to conformational changes in the cricket protein that reduce its electron transport capacity but improve its ability to act as a proton donor, exposing proton-providing residues, namely cysteine [93]. Similarly, studies have been conducted on the effects of thermal processes (e.g., cooking, baking) on the digestibility and cytotoxicity of crickets, and revealed that baked crickets had the highest DH (37.76%) [94]. The hydrolysate of raw, cooked, and baked insects could significantly stimulate or inhibit the proliferation of human skin fibroblast CRL-2522 [94].

Compared with native proteins, the enzymatic modification of proteins is an effective mechanism to improve their functional properties. Hall et al. [17] noticed that cricket protein hydrolysates prepared using Alcalase could exert multiple in vitro bioactive properties. The antioxidant capacity was independent of the DH, while the inhibition value of dipeptidyl peptidase-4 (DPP-IV) of cricket hydrolysates was improved with increased degree of hydrolysis (higher DH, smaller peptide). Patrignani et al. [91] used *Yarrowia lipolytica* and *Debaryomyces hansenii* strains to prepare cricket powder hydrolysates for food products. Compared with the control group, the hydrolysates contained a lower chitin content and higher antibacterial substances (acetic acid, short-chain fatty acids, chitosan, and γ-aminobutyric acid) and health-promoting molecules (arachidonic and linolenic acid, γ-aminobutyric acid, α-amino butyric acid, and β-aminobutyric acid). In addition, the strain increased matrix digestibility (due to proteolytic activity and release of free amino acids) and improved the sensory characteristics of cricket powder. Insect protein has an earthy-like flavor, and even when insect powder is added to some food products, it does not mask the obvious insect taste and reduces the overall acceptability of the food [95]. Due to the release of free amino acids and peptides during enzymolysis, hydrolysates are susceptible to Maillard reaction [95]. Thus, hydrolysis and the Maillard reaction have been found to significantly change the flavor of cricket proteins, rendering them more complex and savory-like taste profiles (27 descriptions of the crickets and 39 descriptors for mealworm protein), and by means of the method of gas chromatography-physical smell identified 38 odors-reactive molecules (1 alcohol, 5 acids, 11 aldehydes, 5 ketones and 16 heterocyclic compounds) [95].

The use of microwave radiation in combination with enzymatic hydrolysis to reduce allergenicity and enhance the bioactivity of peptides has been previously investigated [96,97]. It has thus been shown that the use of microwave radiation could shorten hydrolysis time and increase overall product yield by increasing reaction rates [22,98]. The study by Hall et al. [22] proved that microwave-assisted protein hydrolysis of crickets had the advantage of accelerating hydrolysis reaction, to obtain peptides with enhanced inhibition of DPP-IV and ACE.

### 5.4. Mealworm (Tenebrio molitor)

Mealworms (*T. molitor* larvae) belong to the tenebrionidae family and are generally considered pests because they feed on stored grains [20]. However, these larvae are edible and are receiving global attention as a source of dietary protein. Mealworms grow quickly, and the creatures can be cheaply reproduced and with less burden on the environment [99]. Importantly, yellow mealworms exhibit a strong ability to utilize inorganic wastes [99]. Among edible insects, mealworm larvae are considered as a novel food due to their high nutritional value (high protein, fat, and mineral contents) and their feasibility in the food industry [100,101]. The nutritional quality of mealworms is similar to those of other traditional meat sources [102]. Mealworms contain about 46% protein and 35% fat, and their amino acid composition meets the nutritional needs of humans and animals [20]. Additionally, mealworms have been found to exhibit some bioactivate functions such as anti-hypertension [100], antioxidant (liver protective peptide) [20], and immunomodulatory activity [99], which are all beneficial to human health.

The midgut of mealworm larvae is rich in proteases with metabolic and catalytic activities, and serine endopeptidase with milk-clotting activity was detected [103]. The protease has a high tolerance at low pH and high temperature, and its milk-clotting mechanism was mainly hydrolysis of Lys116-Thr117 bond in κ-casein, which provides a scientific basis for the development of mealworm chymosin [103]. Endogenous phenoloxidase causes undesired browning during the grinding of insects. This enzymatic browning impairs protein qualities, such as solubility, water holding capacity, foaming ability, and digestibility [104]. Studies have demonstrated that endogenous phenoloxidase and protease remain active after mealworm protein extraction. Endogenous proteases can induce a high degree of proteolysis, which may significantly alter the techno-functionality of protein extracts [104].

Mealworms seem very promising as functional ingredients due to their in vitro and in vivo antihypertensive activity [100]. The antihypertensive component of mealworm larval protein was YAN (Tyr-Ala-Asn) tripeptide, and its ACE-inhibition was confirmed, but the low yields of YAN from mealworms did not adequately explain the activity of the whole protein fraction. Moreover, the identification of the other three peptides revealed that the tetrapeptide NIKY (Asn-Ile-Lys-Tyr) exhibited the most promising ACE-inhibition activity (52 μM), which again highlighted the potential of mealworms and paved the way for the development of novel foods [100]. The use of antioxidants is a reasonable strategy to prevent oxidative stress-induced liver injury. Low molecular weight protein hydrolysates (<3 kDa) were prepared from mealworms using various protease hydrolases, demonstrating their direct scavenging activity of reactive oxygen species, possibly protecting liver cells from oxygen species-induced cytotoxicity [20]. Recent studies demonstrated that alcalase hydrolysate (<1 kDa) of mealworm suppresses prooxidant-induced cytotoxicity in AML12 cells by activating cellular antioxidant systems via the Nrf2 pathway. Moreover, the novel hepatoprotective peptides, Ala-Lys-Lys-His-Lys-Glu and Leu-Glu, were also identified and derived from mealworms’ alcalase hydrolysate [20]. Selenium is an essential micronutrient for the maintenance of human and animal health. Selenium deficiency could lead to metabolic syndrome, chronic diseases, and even cancer [105]. It has been previously reported that selenium enrichment could affect the biological activity of hydrolysates [106]. Dong et al. [99] proved that mealworms could accumulate and transform inorganic selenium into its organic form. After hydrolysis using an alkaline protease, the hydrolysate of selenium-enriched larvae had significantly higher radical scavenging and immunoregulatory effects compared to those observed in hydrolysates of regular larvae.

Recently, the functional property of mealworm protein obtained by using novel processing techniques (e.g., ultrasound) were extensively studied. Some reports showed that sonication could substantially be used to improve the functionality of protein, such as the release of bioactive peptides [88]. Ultrasonic treatment may induce different structural and physical modifications of proteins to improve solubilization, thereby contributing to the production of hydrolysates in the case of low-soluble proteins (e.g., protein from insects) [101]. Ultrasound pretreatment modifies the native structure of the protein, and subtilisin hydrolysis reduces the size of the peptide chain, which favors the release of smaller active peptides. Short durations of ultrasonication improved the release of bioactive peptide, but longer durations limited bioactive peptide production. Therefore, ultrasonic pretreatment is mainly applied for a short reaction time to increase bioactive peptides (anti-diabetic peptide source), and could achieve industrial scale-up production [101]. Two edible insects, *Acheta domesticus* and *Tenebrio molitor*, were extracted by ultrasonic-assisted extraction and pressurized liquid extraction using ethanol and hydroethanol as solvents [107]. It was found that all extracts exhibited antioxidant activity, which was associated with the total phenolic compound value and ethanol: water extract had the strongest antioxidant activity. All extracts showed lipase inhibition activity, but mealworms extract and pressurized liquid extraction were the most effective [107].

## 6. Application of Insect Protein

Presently, there is growing interest in the preparation of isolated proteins and/or hydrolysate from insects as functional and natural additives in the industry to improve the nutritional quality and functionality of food products [90]. Moreover, the use of such in food formulations as an alternative protein source could increase consumer acceptability [108]. Consequently, there is a growing body of research on the addition of insect protein/hydrolysates to different food products; thus, there are two obvious lines of research. One of such made an important contribution to the fortification and supplementation of food products (such as cakes, breads, and pasta) to increase their nutritional quality. The other focuses on the reformulation of products aimed at the reduction or replacement of meat or milk proteins [37]. Moreover, Gould et al. [109] found that proteins isolated from mealworms can be used to stabilize oil/water emulsions. They also observed that, compared with whey proteins, a small amount of mealworm proteins was required to produce similar emulsions.

Food fortification is an effective technique to reduce micronutrient malnutrition [110]. It is practiced not only in developing countries with limited food supplies, but also widely in industrialized countries to control vitamin and mineral deficiencies [111]. Wheat flour lacks some essential amino acids, especially lysine and trilysine, affecting its nutritional profile [112]. Therefore, bread is one of the most commonly fortified foods, and the addition of these amino acids through external sources compatible with production technology may help to improve the nutritional value of flour and its derivatives [113]. Among them, innovative additives such as insect flour are used in bread formulations owing to their high protein content, contributing to increased nutritional value of the final product [114].

Several reports reveal that the addition of insect flour to wheat flour enhances the quality and nutritional composition of bread [113,114,115,116]. Furthermore, only 10% of insect flour resulted in an increase in the amino acid score of lysine from around 40% to nearly 70% [114], with an overall increase in essential amino acids [113]. It is also accepted by consumers [114]. The sourdough obtained from cricket powder hydrolysates was rich in health-promoting molecules such as arachidonic acid and linolenic acid, which could be used for innovative bread production with high nutritional and functional value [117]. Using lesser mealworm powder (replacing 10% or 30% of wheat flour) as crunchy snacks (rusks), it was observed that such powder can enhance cereal-based foods with Zn, Fe, and P, and to a lesser extent, in Mg and Ca [118]. Pasini et al. [3] used insect protein powder to replace part of wheat flour to prepare pasta. Pasini and coworkers found that insect powder substantially improved the protein content and quality of the final product with better firmness and adhesiveness of texture, but the cooking loss was larger than that of the control pasta. Nissen et al. [119] used cricket powder to produce gluten-free sourdough breads suitable for people with celiac disease and as a “protein source.” Cricket powder significantly increased the antioxidant activity of bread, providing a high nutritional value including protein and antioxidant properties for gluten-free bread products. The use of cricket powder ultimately conferred to the bakery goods a typical aroma, characterized by a unique bouquet of volatile organic compounds, differently expressed when different types of inocula were applied. The results of sensorial evaluations of cricket-enriched breads recorded similar scores to standard ones.

Insect protein can be also used as a new protein source to partially replace meat or milk components in meat or milk products without affecting their nutritional and technical characteristics. For instance, the emulsified sausage was prepared from defatted mealworm larvae and silkworm pupae powder, which could be used as the suitable raw material for such a product [120]. Tello et al. [121] developed a milk substitute using mealworm larvae as raw material, which has a reduced impact on the environment and a similar nutritional content. The development trial identified a successful method and formulation for developing a basic prototype of “insect milk” consisting of 1.19% crude protein, 5.76% fat, and <1% carbohydrate. Ventanas et al. [122] prepared pizzas with dehydrated mealworm larvae instead of bacon as an animal protein source, and found that consumers provided a positive evaluation of pizzas with mealworm larvae added in terms of vision and sensory assessment. Using mulberry silkworm pupae and locust flours as protein sources in high-energy biscuits (HEB) formulation, it was found that 15% edible powder improved the nutritional properties of HEB, exceeding the minimum expectations and approaching conventional skim milk (control sample) [123]. There are no obvious technical defects in width, thickness, spread ratio, and weight of insecticized biscuits [123]. No matter the appearances, nutritional requirements, sensory evaluations, or microbial contents of biscuits, they were at the same level as the control group [123]. Pellets made from insect flour (*H. illucens*) were prepared by hot melt extrusion [124]. The addition of insect flour affected the gelatinization performance of the formula [124]. Compared with corn flour extrudates, the water solubility index of the extruded solution prepared with insect meal was significantly increased, whereas water and oil absorption capacity were significantly reduced [124]. Thus, this shows the possibility of using insect flour as a protein source during hot melt extrusion.

Furthermore, studies reveal that adding powdered insects to bread not only reduces the digestibility of starch, but also reduces glucose intake in the small intestine, thereby reducing postmeal hyperglycemia [110]. Mealworm larvae have anti-obesity effects in vitro and in vivo, but it remains unknown how the larvae affect appetite regulation in mice with diet-induced obesity. The literature proved that the potential mechanism of mealworm larva extract suppresses appetite by downregulating orexigenic neuropeptides via modulation of hypothalamic mTOR and MAPK signaling pathways [125].

On the other hand, the addition of insect powder to food has some problems, such as biological and chemical risks. Although bacterial pathogens are rarely carried by farmed insects, they could still serve as carriers of contamination, The extent of which depends on insect species, collection method (wild or domestic), rearing environment, and hygiene practices used in insect preparation and processing [126]. Vandeweyer et al. [127] reported that the three main bacterial groups that may pose a risk in insect foods are *Clostridium* spp. (particularly *C. perfingens* and *C. botulinum*), *Bacillus cereus* (*B. cereus* sensu lato (s.l.)), and the non-spore forming species *Staphylococcus aureus*. However, other classical foodborne pathogens such as *Salmonella* spp., *Campylobacter* spp., *Listeria monocytogenes*, or pathogenic *Escherichia coli* are rarely detected, whereas particularly spore-forming bacteria with pathogenic potential such as species of the *Bacillus cereus* group or *Clostridium* species may pose a food safety risk. Frentzel et al. [128] investigated a variety of 73 food products with insect or other arthropod ingredients on the occurrence of potential bacterial pathogens. The results showed that most of the investigated insect or arthropod components of the food met food safety standards for pathogens, but the high incidence of *Bacillus cereus* (s.l.) confirmed the importance of spore-forming bacteria to insect food safety. Therefore, the importance of food processing techniques must be emphasized to ensure that the vegetative cells of pathogens are safely inactivated without promoting the growth of spore formers. Preventive treatments (e.g., blenching, microwave drying, high-pressure, etc.) that reduce the load of this bacterial community are recommended when using insect powder as a food ingredient [129]. In addition to microbial contaminants, chemical contaminants can also pose a hazard to humans. Like other animals, insects may accumulate hazardous chemicals, such as persistent organic pollutants (POPs), flame retardants, and plasticizers [130]. Despite the usage restrictions imposed on POPs, such as polychlorinated biphenyls (PCBs), organochlorine pesticides, and polybrominated diphenyl ethers, due to their persistent, bioaccumulative, and toxic properties, these contaminants are still ubiquitously detected in the environment and in humans [131]. Studies have found that insects may accumulate POPs during feeding or from the wild environment, while organophosphorus flame retardants and plasticizers come from post-harvest industrial treatment and seasoning [132]. Contaminant levels (i.e., flame retardants, PCBs, dichlorodiphenyltrichloroethane (DDT), dioxin compounds, and pesticides) and metals (As, Cd, Co, Cr, Cu, Ni, Pb, Sn, and Zn) were reported to be lower or similar in four edible insects (*Galleria mellonella*, *L. migratoria*, *T. molitor*, and *A. diaperinus*) marketed for human consumption when compared to other animal products (i.e., fish, meat, and eggs) [133]. Thus, the consumption of these insect species, compared to the more commonly consumed animal products, has no additional harm, with reference to these chemical contaminants.

## 7. Conclusions

This review has shown that insect protein is a promising solution for feeding the world’s growing population. Insect proteins have the potential to be produced on an industrial scale in a more sustainable way than traditional protein sources. Edible insects are rich in protein, possessing all the amino acids needed for a healthy diet, and can be processed into protein-rich food ingredients. In this regard, the techno-functional properties of insect proteins are determinative for the selection of possible applications. In addition, foods enriched with insects (i.e., hydrolysates) may prevent a variety of health problems, such as hypertension, hyperlipidemia, hyperglycemia, and cancer (hepatoma and gastric cancer). At the same time, the challenge of allergenic risk in insect protein foods must be addressed to meet the needs of the global society. Consider that the problem of insufficient consumer acceptance relates to cultural aspects and taste and sensory attributes; thus, functionality, cost-effectiveness, sustainability, and consumer safety are not the only aspects that consumers consider when choosing food.

## Figures and Tables

**Figure 1 foods-11-02931-f001:**
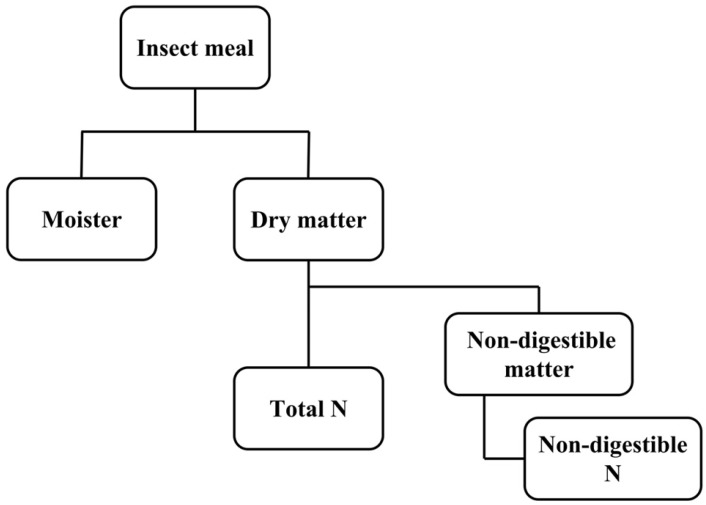
The proposed process to quantify the nutritious proteins from insects: digestible protein content = (Total N–non-digestible N) x conversion factor [31].

**Figure 2 foods-11-02931-f002:**
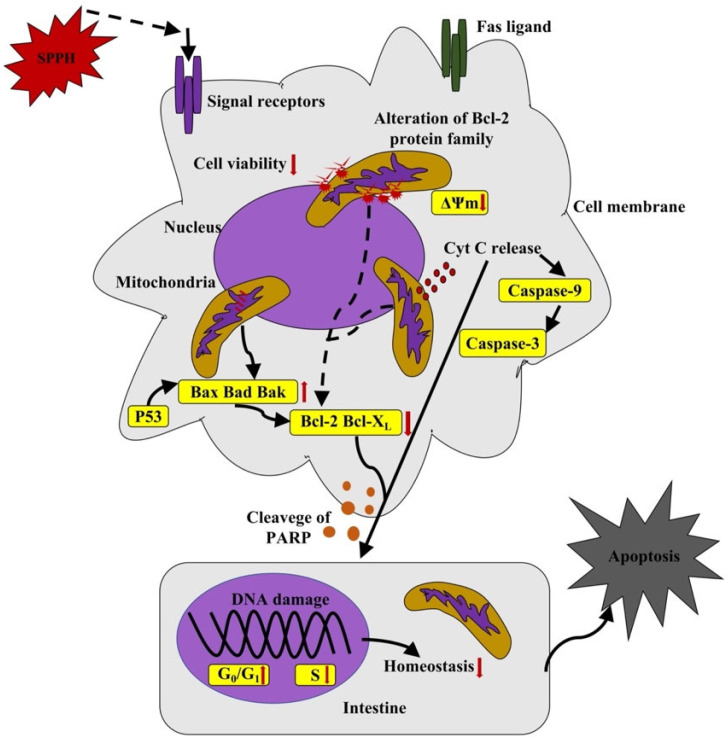
Schematic representation of the mechanism by which SPPH III induces apoptosis in MGC-803 cells [77].

**Table 1 foods-11-02931-t001:** Proximate compositions and micronutrients (minerals and vitamins) of some edible insects (based on dry matter).

		*Bombyx mori*	*Hermetia illucens*	*Acheta domesticus*	*Tenebrio molitor*
Proximate composition	Proteins (%)	48.70–58.00	41.44	64.38–70.75	47.70–49.08
	Fatty acids (%)	30.10–35.00	35.69	18.55–22.8	35.17–37.7
	Fibers (%)	2.00	0.08		5.00–14.96
	Ashes (%)	4.00–8.60	7.87	3.57–5.10	2.36–3.00
	Carbohydrate (%)	1.00	12.85	2.60	7.09–7.10
	Energy (kJ/kg)	23,236.74		19,057.89	22,863.14
Minerals (mg/100 g)	Calcium	158.00	2295.00	132.14–210.00	44.36–47.18
	Potassium		478.00	1126.62	761.54–895.01
	Magnesium	207.00	220.00	80.00–109.42	210.24–221.54
	Phosphor	474.00	547.00	780.00–957.79	697.44–748.03
	Sodium		204.00	435.06	125.38–140.94
	Iron	26.00	27.00	6.27–11.23	5.41–5.51
	Zinc	23.00	6.90	18.64–21.79	11.41–13.65
	Manganese	0.71	13.06	2.97–3.73	0.92–1.36
	Copper	0.15	1.12	0.85–2.01	1.60–1.64
	Selenium	0.15	0.07	0.60	0.03–0.07
Vitamins	Retinol (μg/100 g)		118.00	24.33	
	α-Tocopherol (IU/kg)		80.39	63.96–81.00	
	Ascorbic acid (mg/100 g)			9.74	3.15–6.15
	Thiamin (mg/100 g)			0.13	0.31–0.63
	Riboflavin (mg/100 g)			11.07	0.41–2.13
	Niacin (mg/100 g)	0.95		12.59	10.59–10.68
	Pantothenic acid (mg/100 g)			7.47	3.72–6.88
	Biotin (μg/100 g)			55.19	78.74–94.87
	Folic acid (mg/100 g)			0.49	0.30–0.41
Reference		[11]	[12]	[11]	[9]

**Table 2 foods-11-02931-t002:** Essential amino acid contents of edible insects as compared with other food proteins (egg whites and soybeans) and with the FAO standard protein (mg/gprotein).

Essential Amino Acids (mg/g_protein_)	*Bombyx mori*	*Hermetia illucens*	*Acheta domesticus*	*Tenebrio molitor*	Egg White	Soybean	Amino Acids Required in Human Nutrition (FAO)
His	25.8–29.5	33.0	21.0	35.3–37.9	22.0	25.0	15.0
Thr	28.4–31.2	42.0	35.0	34.8–40.8	47.0	38.0	23.0
Val	39.8–40.9	66.0	60.0	66.3–69.0	68.0	43.0	39.0
Lys	47.3–50.0	65.0	56.0	60.9–64.9	70.0	63.0	45.0
Ile	32.3–33.0	41.0	42.0	46.7–49.4	53.0	47.0	30.0
Leu	48.9–52.7	75.0	73.0	77.7–82.2	88.0	85.0	59.0
Phe	28.4–29.0	36.0	33.0	40.8–43.7			
Trp	6.8–7.5	9.0	6.0	9.2–10.3	14.0	11.0	6.0
Cys	8.6–9.1	30.0	21.0	8.2–10.9			
Phe + Tyr	60.2–62.5	110.0	74.0	118.5–123.6	91.0	97.0	38.0
Cys + Met	21.6–22.6	47.0	36.0	22.3–30.5	66.0	68.0	22.0
Reference	[9]	[13]	[9]	[9]	[13]	[13]	[13]

Note: His, histidine; Thr, threonine; Val, valine; Lys, Lysine; Ile, Isoleucine; Leu, leucine; Phe, phenylalanine; Trp, tryptophan; Cys, cysteine; Tyr, tyrosine; Met, methionine.

## Data Availability

Not applicable.

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
