# Peer review of "Recent Insight on Edible Insect Protein: Extraction, Functional Properties, Allergenicity, Bioactivity, and Applications"

_foods, 2022, doi:10.3390/foods11192931_

Round 1

Reviewer 1 Report

The article has too many disconnected ideas. For instance, in the paragraph entitled “Extraction of insect protein” author only mentioned “Alkaline extraction and isoelectric precipitation” that is the most traditional one, listed the steps and started talking about other studies without even mentioning other possibilities and started talking about “main unit operation after harvesting insect is the devitalization technique employed,” and “killing options” which in not at all linked to the extraction process itself but with the influence it causes in the quality of the product. In that sense the article has no flow and under each section author are not specific at all and just list diverse results on different things found in other articles. For that reason, I do not think the article is suitable for publication as it does not have a clear objective. As a suggestion for future author can divide the article in sections such as “protein extraction processes”, “Influence of process X on  A, B, C (e.g. nutritional, functional, sensory)”. In that way the ideas would be better connected instead of under one section having information of many diverse and unliked topics in the same section.

Also, the texts repeat itself throughout the article. For instance, in section 3 author again talk about population growth and low acceptance of insect (Lines 217- 221). This info was already mentioned and there is no need to repeat it after each section.

In addition, author make very specific statements as if this is the only alternative. For instance, In the abstract author mention the need to find new sustainable and natural protein resources which is true, but they only mention it needs to be from invertebrates, however it can also be plant-based. Therefore, author should not make such a specific statement as the need is not only from invertebrates but from any source, animal or plant-based, which are more sustainable then the ones currently being used.

Moreover, in the last years there has been numerous high-quality articles in the literature in the same thematic. Authors should make a compelling statement as to why is there a need for another article with the same content? What is the added value from this review? Would it be better perhaps to focus on one or two specific sections instead of trying to address everything?

Examples of other review articles (and there are many others which I did not list here):

https://www.sciencedirect.com/science/article/pii/S1359029421000923

https://www.sciencedirect.com/science/article/pii/S0260877421002855

https://onlinelibrary.wiley.com/doi/full/10.1002/mnfr.201600520?casa_token=oT-gYyyyJYgAAAAA%3A5-7e9lw5pxj7YhHXuk38njmC426VwsFgPqP88hbQM-iMXs0AGhcvvD5kUzbT-Vv4Q_7UJOnrZXtCKpv7Bg

https://ift.onlinelibrary.wiley.com/doi/abs/10.1111/1541-4337.12463

https://www.sciencedirect.com/science/article/pii/S030881461932165X?casa_token=wE37z2otPQUAAAAA:KvVbc9IBJ5TBzm54_cT3PGggam96g4re2AfeZ9XdSqkobpCfhId3V3k1r4gxZXjs8J219ltc2Xt-

https://www.sciencedirect.com/science/article/pii/S0924224417306702?casa_token=XbRC1wC19x4AAAAA:kU2eez452KqJNcbp947Xer_HD0Tv4EpimPjVPsbqpEfpB7VBjVnf_rmI2r4nT3_mG_j9pS_a7Ao9

Other minor comments:

Line 32: Avoid using the word “today” in a scientific article. Be specific and mention the year

Line 36: Punctuation

Line 40: Why apart from? If not for food what would be the purpose of insect farms? Be specific.

Line 51: Insect is also animal; author should rethink the use of their terminology

Line 73: Making a powder is not a method on its on but rather a process…

Reviewer 2 Report

The manuscriptRecent insight on edible insect protein: extraction, functional properties, allergenicity, bioactivity and applications” is generally interesting and covers an important topic, however, I think there is a number of aspects which should be addressed before it could be accepted for publication. Below I have provided some specific comments & suggestions for the Authors:

‘Technical’ aspects:

First letters of all words in the title should be capital.

Section and subsection titles – all words should begin with capital letters (except for “and”, “of” etc.).

Abstract:

Many sentences of the abstract need careful revision. Examples:

Line 16: human population

Line 17-18: Sentence needs revising to make it clear that it is necessary to look for alternative protein sources, and insects are among the options.

Line 18 “(…)Compared with animal/ plant proteins, insect proteins…” – in fact insects are animals, so this also needs revising.

Line 18-19:  It’s incorrect to say that insect proteins have advantages in terms of protein content. And this needs revising as well.

Introduction:

Most importantly: Lines 33, 37, 50, 81, 82, 142, 218… (and more): The authors refer to another review on the same subject, published in Food Chemistry in 2020. Instead they should analyse the most current original research of the topic and cite relevant articles.

Lines 32-33: human population.

Line 50: pork and chicken should not be in brackets, but possibly after comma.

Table 2: Full names of amino acids should be provided (either in the table or in the table footnote). Since data for egg and soybean are given for comparison, this should be also mentioned in the table title. For the column „Amino acids required in human Nutrition” unit should be specified in the table.

Line 71: What type of bias do the authors mean? Unclear.

Line 86: Space between words is missing.

2. Extraction of insect protein:

Fig 1: Should be “Figure 1” (the same applies to Figure 2). Moreover, “Non-digestible N” is in fact part of “Total N” – this should be corrected in the figure.

5. Bioactivity of insect proteins:

The title of this section could be changed, since it provides various information about particular insect species (and not only information about the “bioactivity of their proteins”).

Line 371: Cancer = tumors. I suggest to keep cancer only (and to write: “certain types of cancer”).

Line 371: It’s unnecessary to mention Latin name again.

Lines 373-376: These impacts should be grouped logically (e.g.

7. Conclusion:

Title of the section: change to “Conclusions”.

Line 747: Diabetes is mentioned for the first time here. If there is relevant research about this, it should be presented in one of the previous sections, and not mentioned suddenly in the conclusions, without giving any reference.

Line 750-753: I would suggest to revise this last concluding sentence to take into consideration the problem of lack of consumers’ acceptance related to cultural aspects and taste/sensory attributes. Functionality, cost-effectiveness, sustainability, and consumer safety are not the only aspects which consumers take into consideration when making their food choices.

General comment:

More information (maybe a separate section) about the microbiological and chemical safety of edible insects would be nice. At the moment there is rather limited information about these aspects in sections: 4. Allergenic risks of insect protein, and 6. Application of insect protein.

References:

Following the requirements of Foods (MDPI), all names of the journals in the listed references should be abbreviated.

Round 2

Reviewer 1 Report

The article is improved; however, it still lacks a better organization of the information. For instance, Lines 131-137 is more related to functionality not to the extraction processes itself. Therefore, I still recommend that author connect more the ideas and make a better division of article content. Authors should make sure that under each section they are ONLY sticking to the section content and not mixing it all. In the abstract author say, “introduce in detail the latest advances in extraction process”. However, authors are ONLY mentioning isoelectric precipitation, how about the other methods? If article aim it to detail the “latest advances”, shouldn’t ALL latest extraction processes be mentioned? How about enzyme? Salta extraction? Membrane filtration? Again, this paragraph talks more about functionality and nutritional than the extraction process itself. Moreover, the use of the term “meat” protein is weird, I suggest authors are more specific. Also, there are still many sentences that are completely out of place/section (e.g. line181-184), what does this have to do with the two option you just mentioned? Again, it talks about functionality. For all those reasons I recommend author organize the article better and make sure what is the real aim of the paper!

Other minor comments:

Line 17: new “and” sustainable

Line 18-20 and 69-72: This sentence is not fully correct. Meat protein in an awkward term, most common is animal. Plus, both animal and plant also have “high quality” of all that was mentioned so “comparing” them in such a vague way is not OK.

Table 1 is misleading as minerals and vitamins (micronutrients) are also “nutrients”. I suggest authors re-think their classification on insect composition

Line 98: This sentence is not linking to what was previously said. Wouldn’t it be Hence, processing insects in the powder form in its various forms such as….

Line 130: Not only this but also simply as flour, without any processing

Line 181-184: Out of place, disconnected

Line 1043: There were…
